# The Relationship between Obesity and Physical Activity of Children in the Spotlight of Their Parents’ Excessive Body Weight

**DOI:** 10.3390/ijerph17238737

**Published:** 2020-11-24

**Authors:** Erik Sigmund, Dagmar Sigmundová

**Affiliations:** Faculty of Physical Culture, Palacký University Olomouc, 77147 Olomouc, Czech Republic; dagmar.sigmundova@upol.cz

**Keywords:** organized leisure-time physical activity, overweight, obesity, parent-child dyads, step counts

## Abstract

Background: The study reveals the relationships between daily physical activity (PA) and the prevalence of obesity in family members separated according to the participation of their offspring in organized leisure-time physical activity (OLTPA), and answers the question of whether the participation of children in OLTPA is associated with a lower prevalence of obesity in offspring with respect to parental PA and body weight level. Methods: The cross-sectional study included 1493 parent-child dyads (915/578 mother/father-child aged 4–16 years) from Czechia selected by two-stage stratified random sampling with complete data on body weight status and weekly PA gathered over a regular school week between 2013 and 2019. Results: The children who participated in OLTPA ≥ three times a week had a significantly lower (*p* < 0.005) prevalence of obesity than the children without participation in OLTPA (5.0% vs. 11.1%). Even in the case of overweight/obese mothers/fathers, the children with OLTPA ≥ three times a week had a significantly lower (*p* < 0.002) prevalence of obesity than the children without OLTPA (6.7%/4.2% vs. 14.9%/10.7%). Conclusions: The cumulative effect of regular participation in OLTPA and a child’s own PA is a stronger alleviator of children’s obesity than their parents’ risk of overweight/obesity.

## 1. Introduction

The regular participation of children and adolescents in organized leisure-time physical activities (OLTPA), including individual and team sports, contributes to or enhances several health-related benefits [1,2,3,4,5]. In general, adolescents who are regularly involved in OLTPA show better mental health (e.g., psychological resilience [2], internalizing social and emotional problems [3], lower depressive symptoms [4]), less frequent risky behavior (smoking [4,5], drunkenness, and skipping school [5]). Although the relationship between OLTPA and obesity in youth is less clear [6], there are indications that regular participation in OLTPA plays a role in reducing obesity [7,8,9,10,11]. Engagement in OLTPA as early as kindergarten and the first grade of primary school are associated with smaller increases in the Body Mass Index (BMI) during the adiposity rebound period of childhood [7]. Subsequently, regular participation in OLTPA at least once a week during one’s primary school years is significantly associated with a decreased risk of overweight/obesity [8,9] and a lower accumulation of excess body fat [10,11]. Nevertheless, the participation of children and adolescents in OLTPA and overall physical activity (PA) is mediated by parents [12,13,14].

Since parents (guardians) are essential in caring for children and creating their home environment, the parents have been described as “gatekeepers” of children’s health-related behaviors by determining what activities children do, as well as what resources and access they have at their disposal [15]. In the context of the overall PA and OLTPA of children and adolescents, existing parental support mechanisms include: motivational (the provision of verbal/non-verbal prompts to engage in the behavior of interest, validation, and affirmation of involvement of performance by participating in the behavior), instrumental (the provision of tangible aid and/or services), conditional (being directly involved in the activity with the child or in its vicinity), and regulatory (enforcing rules and/or setting limits) categories [16,17]. It has been repeatedly confirmed that physically active parents bring up more physically active children [14,17] and positive parental support meaningfully contributes to a higher PA of the offspring [1,12,16,17]. However, mothers and fathers influence their children’s PA differently [16] and it seems to be stronger in parent-child pairs of the same gender [14,18]. It turns out that the closest relationship between the PA of parents and that of their children is strongest when the child is eight years of age, with a gradual weakening until the children are 16 years of age. In addition, with the increasing age of children, gender-specific PA parent-child relationships occur [18].

Similar to the linked participation of children in OLTPA with the support of parents, the relationship between the level of bodyweight of parents and their offspring have been repeatedly confirmed [19,20,21,22]. Children who have one obese parent have an increased risk of being overweight or obese [19,20], which is exacerbated if both parents become overweight/obese [21,22]. Although the findings are inconsistent, children with more physically active parents appear to be less likely to be overweight and obese [13,23]. The relationships between the bodyweight levels of parents and their offspring have been examined with regard to different family correlates (gender, age, level of education, PA, screen time, and employment of parents [19,21,22,23], socioeconomic status, and structure or income of the family [13,19,20,21,22]), but rarely with regard to children’s participation in OLTPA [23].

There is no doubt that the worldwide prevalence of childhood and adolescent obesity is a serious health problem [24] and its incidence has not been satisfactorily reduced globally [25,26]. Childhood obesity is the consequence of the interaction among a complex cluster of environmental, genetic, and psychosocial determinants that often result in excessive caloric intake, as well as insufficient PA [27]. Additionally, the solution to childhood obesity is not simply achieved by reducing the calories taken and increasing PA while the failure of obesity reduction programs simply cannot simply be seen as a lack of self-control or low program adherence [27]. Unfortunately, most of the existing child obesity reduction and prevention programs have all had very small effects, with high heterogeneity [26,28,29,30]. One of the reasons for the low level of effectiveness of interventions is their pilot verification in optimal conditions, which very rarely prevail during the transfer to the everyday living conditions of the real world [28,29]. The importance of pilot verification of intervention programs regarding a possible generalizable bias has been re-emphasized [30]. From a public health perspective, it is therefore still desirable to seek and shed light on relevant correlates that prevent the rise of childhood obesity in everyday living conditions. Previous studies point to the importance of the well-being and enhanced potential of children’s health including regular participation in organized leisure-time activities [5,31], but the relationship between children’s regular participation in OLTPA, as well as the occurrence of their obesity in the spotlight of PA and parental body mass levels, has not yet been analysed.

The participation of children and adolescents in OLTPA in Czechia is the most widespread form of spending on organized leisure time for young people (≥74% for boys and ≥63% for girls [5,31]). However, the trend in the prevalence of obesity in Czech youngsters is still rising [32], so it is appropriate to examine how the regular participation of children and adolescents in OLTPA is associated with obesity in the context of parental bodyweight levels and PA. To address this research gap, the aim of this study is to determine the relationships between overall daily PA and the prevalence of obesity in family members separated according to the participation of their offspring in OLTPA, as well as to answer the question of whether the regular participation of children in OLTPA is related with a lower prevalence of obesity in offspring with respect to parental PA and bodyweight level. In this study, we will test the hypothesis of whether there is a difference in the prevalence of obesity in offspring without and with regular participation in OLTPA (three or more times per week) in the case of excessive body weight in their parents.

## 2. Materials and Methods

### 2.1. Ethics

The study design, all procedures, and the measurement, as well as the method used for feedback, were approved by the Ethics Committee of the Faculty of Physical Culture, Palacký University, Olomouc separately for families with preschool children (ref. No.: 57/2014 on 21 December 2014), families with 6–11-year-old children (ref. No.: 20/2012 on 12 December 2012), and families with 12–15-year-old adolescents (ref. No.: 14/2018 on 21 February 2018). The parents’ written consent was obtained prior to the start of the data collection (Scheme 1). The parents of the children gave their consent to participation in this study. Participation in the project was voluntary and without financial incentives.

### 2.2. Participants and Inclusion/Exclusion Criteria

The method used to select the respondents has been described in detail in previous studies [18,23]. Participants were recruited by means of a two-stage stratified random sampling. In the first stage, nine out of 14 administrative regions, three of each in the lowest, middle, and highest terciles for gross domestic product in Czechia, were randomly selected. In the second stage, seven public kindergartens located in urban areas and three in rural locations were selected, while there were 36 public primary schools located in urban areas and 15 in rural locations which were randomly selected [23]. A total of 3540 families were addressed in writing with an invitation to participate in the cross-sectional study, of whom 65.3% agreed to take part in the research (written informed consent received). The participating children and their parents/guardians were predominantly white Caucasian (>98%), which is representative of the ethnic demographics of Czechia [33]. Scheme 1 provides a detailed flowchart of the inclusion of the participants in the study. The family dyads consisted of a father-child or mother-child pair who shared living quarters. To be included in the study, at least one dyad (either father-child or mother-child) per family had to provide informed consent. The final dataset consists of 1493 family dyads (915 mother-child and 578 father-child) with complete data on the weight status and ambulatory PA of family members monitored with a Yamax pedometer during a regular school/work week during the spring and autumn between 2013 and 2019 (Scheme 1). The basic somatic characteristics of the participants included in the study are presented in Table 1.

### 2.3. Procedures and Measurement

Prior to the information meeting being held in the kindergartens/schools that agreed to participate in the study, the kindergartens/schools received pedometers for familiarization between the children and teachers. In the initial stage of the study, information meetings were held to describe the process of the research, ways of handling the pedometer, and the way of recording data in the family logbook with all the participants in the kindergartens/schools. Upon providing written informed consent, each family received a self-monitoring package including: (1) a letter describing the study design and the ethical approval, (2) a family logbook for recording the anthropometric and step counts (SC) and OLTPA data of all family members, (3) an illustrated instruction leaflet for home measurement of the body weight and height of the family participants, (4) Yamax Digiwalker SW-200 (Yamax Corporation, Tokyo, Japan) pedometers with illustrated instructions describing how to operate them, and (5) an explanatory letter to the teachers/coaches about the study explaining why a pedometer is worn by the children during lessons/training.

The parents were asked to fill in the anthropometric data (date of birth of the child/children, age of parents, gender, and body weight/height (with 0.5-kg/cm accuracy)) of all the participating family members in the family logbook before the start of the PA monitoring. The parents were thoroughly instructed on how to measure their own body height and weight, as well as the height and weight of their offspring, according to the illustrated instruction leaflets for home measurement. Parental measurement of the body weight and height of their preschool children and schoolchildren at home is a sufficiently valid method to identify overweight/obesity according to the calculated BMI compared with objective or laboratory/researcher measurement [37,38,39].

The all-day PA of all the family dyads was monitored with an identical unsealed Yamax Digiwalker SW-200 pedometer and quantified by the SC during waking hours. The participants were asked to wear the pedometer attached to their right hip for eight consecutive days for at least eight hours per day and record their daily SC and active participation in OLTPA in the family logbook that was provided [40]. The participants were instructed to wear the pedometer throughout the whole day (during their journey to school/work and during classes and breaks, as well as during participation in OLTPA) except when dressing, performing personal hygiene, and showering/bathing. Each morning, the parents and their school-aged children recorded the time of attachment and SC value; each evening they recorded the time of removal and SC value. In the case of preschool children, this data was proxy-reported by their parents. The SC data from the first day of PA monitoring was not included in the final analyses because of the novelty of wearing the pedometer, which might have affected the level of the participants’ PA [41,42]. The mean PA monitoring time (excluding the first day) was 6.8 ± 0.68 days with an average daily pedometer wear time of 819 ± 61 min (13.7 ± 1.1 hours). The elimination of seasonal differences was sought by choosing the spring and autumn months in weeks without excessive examinations in schools and without multi-day school holidays as well as public holidays. Although there are more accurate objective tools for measuring PA, the Yamax Digiwalker SW-200 pedometer is an unobtrusive, simple, valid, and reliable quantifier of all-day ambulatory PA across a wide population of children, adolescents [41,43], and adults [44] designed for an analysis of the relationship between daily SC, as well as health outcomes [45].

### 2.4. Data Processing and Statistical Analysis

All the data processing and statistical analyses were performed in the Statistical Package for the Social Sciences (SPSS) for Windows v.22 software (IBM Corp. Released 2013. Armonk, NY, USA). The chronological age of all family members was calculated from the date of birth until the starting day of the monitoring of PA. The BMI was computed as the body weight (kg) divided by body height (m) squared. In accordance with previous studies [18,23,46], the age-specific cut-off points were used to define the prevalence of overweight/obesity [34,35]. Overweight or obesity in children represents a BMI from the 85th to 97th percentile or greater than the 97th percentile of the WHO growth charts [34,35]. Overweight and obesity in parents represents a BMI from 25 kg/m^2^ to 29.9 kg/m^2^ and greater than or equal to 30 kg/m^2^, respectively [36]. Regular participation in OLTPA was defined in accordance with the previous study as three or more times per week [40]. Chi-square (*χ*^2^) tests were performed to compare the prevalence of obesity between children with regular participation in OLTPA (three or more times a week) and children without participation in OLTPA. The daily SC variable represented the mean difference between the morning (pedometer turned on) and evening (pedometer turned off) SC on the days of the week that were monitored. Daily SC values above 30,000 and below 1000 were truncated to these recommended limit values, respectively [42,47], and included in the analyses. The daily SC recommendation was set at a value of ≥13,000/≥11,000 steps/day for 5–12-year-old sons/daughters and ≥10,000 steps/day for 12–16-year-old adolescents [48] and adults [49]. The normality of the SC variable distribution was tested using the Shapiro-Wilk and Kolmogorov-Smirnov test. The Mann-Whitney U test was used to compare the SC variable between children with regular participation in OLTPA and children without participation in OLTPA, as well as the parental SC separated according to the weekly participation of their offspring in OLTPA. Binary logistic regression analyses (Enter method) were performed to reveal whether the regular participation of children in OLTPA is associated with a lower chance of the prevalence of obesity in offspring with respect to the level of PA and bodyweight of the parents. Because of previous differences in the relationships between PA of mothers/fathers and their children [18], regression models were calculated separately for mother/father-daughter/son dyads. An ordinary single-level regression was used because the initial analyses were not significantly altered by the clustering of the PA data by school. The regression parameters were based on the odds ratio (OR) with a 95% confidence interval (CI). An alpha level of 5% was set at a minimum for all the statistical procedures.

## 3. Results

### 3.1. Daily SC of Family Members Separated by the Participation in OLTPA of Offspring

Regular participation in OLTPA was found in 22.8% of the girls and 27.4% of the boys, and 39.2% of the girls and 40.1% of the boys recorded no participation in any OLTPA (Figure 1). A different PA pattern in the daily SC was revealed between the girls and boys separated by the frequency of weekly participation in OLTPA. While a significant increase in the daily SC with more frequent participation in OLTPA was observed in the girls, in the boys the median value of the daily SC remained almost the same, regardless of the frequency of participation in OLTPA (Figure 1). Following the previous result, a significant difference in the median value of the daily SC (difference = 1350 steps/day, *p* < 0.05 based on the Mann-Whitney U test) was found between the girls and boys without participation in OLTPA, which was, however, erased with a higher frequency of children’s participation in OLTPA (Figure 1). Parents divided according to the frequency of their children’s participation in OLTPA did not show significant differences in their daily SC.

### 3.2. Prevalence of Obesity of Family Members Separated by the Participation in OLTPA of Offspring

Offspring who participated in OLTPA three or more times a week had a lower prevalence of obesity than offspring without participation in OLTPA. A significantly (*p* < 0.005) lower prevalence of obesity in children with regular participation in OLTPA (three or more times a week) than in children without participation in OLTPA was revealed in the case of sons (5.3% vs. 14.5%) as well as sons and daughters combined (5.0% vs. 11.1%) (Figure 2).

Even in the case of parents (mothers/fathers) with excessive body weight, children with regular participation in OLTPA had a significantly lower (*p* < 0.002) prevalence of obesity than children without participation in OLTPA (6.7%/4.2% vs. 14.9%/10.7%) (Figure 3). The above-mentioned difference in the prevalence of obesity between children with regular and no participation in OLTPA is 6.5 percentage points for overweight/obese fathers and 8.2 percentage points for overweight/obese mothers (Figure 3).

### 3.3. Association between Child/Parent lifestyle indicators and Obesity of Offspring

The relationships between children’s overall daily PA, participation in OLTPA, parental PA, and body weight level with the chances of obesity in their offspring are presented in gender-separated binary logistic regression models in Table 2. Achieving the recommended daily SC in sons significantly reduces their odds of the occurrence of obesity in both mother-child and father-child models. However, among the boys who are obese, only one-fifth meet the recommended overall daily SC, while among the girls who are obese, the daily recommended SC was reached by as many as 40% of them. Regular participation of the children in OLTPA (three or more times a week) reduced the chance of their obesity in all the calculated regression models but significantly only in sons in the father-child model (Table 2).

The achievement of the recommended daily SC by parents or their participation in OLTPA did not have a significant effect on the chance of obesity occurring in their offspring in any of the regression models. However, parental excessive body weight was positively associated with a higher chance of obesity in their offspring, but this association is significant only in the mother-daughter/son regression models (Table 2). The odds of childhood obesity were lower (OR = 0.36, 95% CI = 0.20–0.65, *p* < 0.001) if children (regardless of the child’s gender) achieved the recommended daily SC and were higher (OR = 2.59, 95% CI = 1.56–4.29, *p* < 0.001) if their mothers were overweight/obese (mother-child regression model). The odds of childhood obesity were lower (OR = 0.45, 95% CI = 0.22–0.92, *p* < 0.03) if children (regardless of the child’s gender) achieved the recommended daily SC or if children participated regularly in OLTPA (three or more times a week) (OR = 0.37, 95% CI = 0.13–0.99, *p* = 0.05) (father-child regression model).

## 4. Discussion

The key finding of the study is the confirmation of the hypothesis concerning the significant difference in the prevalence of obesity in offspring with and without regular participation in OLTPA (three or more times per week) in the case of excessive body weight in their parents. However, regular participation of children in OLTPA is not significantly associated with a lower likelihood of obesity among sons or daughters, except for father-son dyads. In the case of sons, their likelihood of obesity is reduced by reaching the recommended daily SC, regardless of mother/father-son dyads. Children, regardless of gender, are more likely to be obese if their mother is overweight/obese. In Czechia [23], as well as the surrounding European countries [50] and on a global scale [24], the prevalence of youth obesity is increasing, especially for families with low and medium socioeconomic status [23,50]. Therefore, from a public health perspective, it is necessary to look for and make visible the existing correlates reducing the chance of childhood obesity in everyday real life.

In our representative sample of Czech families, participation in OLTPA was found in approximately 60% of the children and adolescents; this is in accordance with previous studies [5,31]. OLTPA is the most widespread type of structured leisure time for young Czech people. Although the positive benefits of OLTPA for the physical and mental health of adolescents have been demonstrated [1,2,3,4,5], its relationship to the prevalence of childhood obesity is less frequently analysed [7,8,9,10,11]. There is still a research gap that remains and which we need to shed light on to try to find the relationship between the frequency of child and adolescent participation in OLTPA, as well as the prevalence of their obesity with respect to the parental body weight and PA level.

The amount of all-day overall PA, quantified by a pedometer in children divided according to their participation in OLTPA, indicates differences between girls and boys. As in previous studies [47,48,51], higher PA was found in the boys than in the girls but only in children without OLTPA. More frequent weekly participation in OLTPA contributes to the higher all-day PA of children [52], but in our case, a significant increase in the daily SC with increasing participation in OLTPA was recorded only in the girls. While for the boys without OLTPA and the boys participating in OLTPA three or more times weekly, the difference in the week-long sum was negligible, at 2319 steps, for the girls this difference represented a total of 16,233 steps per week. An explanation is offered as we believe that boys may engage in unorganized PA in their free time more spontaneously than girls. However, it should be noted that the pedometer is limited when it comes to detecting vigorous PA, which is more typical of boys than girls [53]. The participation of children in OLTPA is not the source of the difference in the overall daily SC for either mothers or fathers.

In contrast to PA, the prevalence of obesity in individual family members is meaningfully differentiated by separation according to children’s participation in OLTPA. A gradual decrease in the prevalence of obesity with the increasing participation of children in OLTPA is evident in daughters, sons, and mothers. Many studies point to a positive effect or association between active participation in OLTPA, as well as the reduced adiposity of children and adolescents [7,8,9,10,11,32]. However, fewer studies take into account the weekly frequency of participation in OLTPA [7,9,11] or even the mediating influence of parents [23]. It has been confirmed that participation in OLTPA more than once per week is more effective at a lower level of adiposity than participation in OLTPA once or not at all [7,9,11]. Just as other studies have pointed out [9,11], the lowest prevalence of obesity was found in children who participated in OLTPA three or more times a week, and in addition, the lowest prevalence of obesity was found in parents of children with regular (three or more times a week) participation in OLTPA too. A parallel can be found in the global recommendation on PA for the health of children and adolescents aged 5–17 years, which also mentions the implementation of activities of vigorous-intensity, which includes strengthening muscles and bones at least three times per week [54]. Additionally, it is vigorous-intensity PA and activities aimed at strengthening muscles as well as bones that are also regularly implemented as part of OLTPA [40,55].

From a public health perspective, the finding of the significantly lowest prevalence of obesity in children with regular (three or more times a week) participation in OLTPA in the case of overweight/obese parents is more serious than the finding of a simple lowest prevalence of obesity in children with regular participation in OLTPA. Although Erkelenz et al. [13] did not find a direct relationship between parents’ and children’s PA, children with at least one parent active in PA showed higher levels of participation in OLTPA and were less likely to be overweight/obese. Existing evidence points to the small to moderate effect of parental involvement in PA promotion and multicomponent (including regular PA) programs aimed at reducing childhood obesity [56,57,58]. Although the current systematic analysis confirms the clear positive relationship between parent and child PA, regardless of the age of the child, the gender of the parent-child dyad, and the type of PA, it also states that this relationship is weak and allows for the action of other mechanisms which, through parents, influence the PA of their offspring [59]. Such effective mechanisms include parental modeling, especially for families with low socioeconomic status or poorly-educated parents [1]. It is on the axis between the PA of children and parents that OLTPA mediated by parents could lie [1,60]. In accordance with thematically similar studies [1,12,13,14,61], we agree with the essential parental modeling role in shaping the active lifestyle of their offspring, which may result in a reduced incidence of childhood obesity despite parental overweight/obesity status.

Among the strengths of this study is the large sample of parent-child dyads for whom the strict inclusion criteria of pedometer-measured PA (numbers of days and daily monitoring periods) were applied. Wearing a pedometer for at least eight hours, including school and leisure time, allows for the relevant capture of overall daily PA. Nonetheless, the study has some serious limitations which need to be mentioned. The observed parent-child dyads did not have to be paired along a biological line, as the participating family dyads were defined as a parent/guardian and a child who shared living quarters. The socioeconomic status of the families was determined only in the sub-group of families with adolescents aged 12–15 years, so it was not included in this study. However, the results showed a non-significant relationship between the socioeconomic status of families and daily SC or the frequency of participation of their 12–15-year-old offspring in OLTPA [62]. Another possible limit is the consideration that young people who are not obese are more likely to self-select into OLTPA than obese individuals. Next, this study used one of the most common pedometers, the Yamax Digiwalker SW-200. Unlike accelerometers, it is not designed to collect non-locomotory forms of movement, intensity, or bouts of activity [38,40,63,64]. However, counting steps is still the fundamental unit of human locomotion and daily SC has strong associations with physical health variables [45]. Given the positive levels of inter- and intra-pedometer reliability, pedometers are effective in large-scale studies as a valid determinant of PA levels among children and adolescents [65]. In the case of OLTPA, which is also characterized by the implementation of vigorous-intensity PA, concerns about a possible underestimation of the total amount of PA are justified. The design of this study was cross-sectional, and the associations revealed do not confirm causality, although they are significant. Regardless of the above-mentioned limitations, the results presented here can serve as important health-related references for researchers and practitioners, as well as families and public health stakeholders, because the participation of children, as well as adolescents, in OLTPA is one of the most widespread ways in which young people spend their leisure time in Czechia [5,31] which can contribute to better mental health [66] and lower involvement in unhealthy lifestyle habits [67].

## 5. Conclusions

The lowest prevalence of obesity was observed in offspring with regular participation in OLTPA (three or more times per week) compared with offspring without participation in OLTPA, even in the case of excessive body weight in their parents. However, regular participation of children in OLTPA was not significantly associated with a lower likelihood of obesity among sons or daughters, except for father-son dyads. Children, regardless of gender, were more likely to be obese if their mothers were overweight/obese. It turns out that a satisfactory explanation for the prevalence of childhood obesity should include other energy-related behavior, lifestyle, and environmental variables in addition to anthropometric and PA data of parents.

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
