# Peer review of "The Relationship between Obesity and Physical Activity of Children in the Spotlight of Their Parents’ Excessive Body Weight"

_ijerph, 2020, doi:10.3390/ijerph17238737_

Round 1
Reviewer 1 Report
This manuscript described the relationship between obesity, step counts, and presumably self-reported time spent in OLTPA in youth and parent dyads. Below are comments to improve the manuscript:
Introduction
The authors present at well-reviewed justification for their study. However, in places the wording is awkward, and should be edited for correct grammar and word choice.
Methods
Lines 125-132: were parents and children instructed to take anthropometric measurements fasting?
Yamax pedometer: how do the authors think knowledge of step count influenced their findings by providing direct participant feedback? Aside from removing the first day, were only whole-days included in analyses? Were there any criteria for number of whole days used? Also, later in the manuscript, the authors say they recommended a step count, however this procedure is not well-described. Was this an intervention study? Did they tell both the parents and children to increase their step counts? More information is needed about how this was communicated.
How was OLTPA measured? Was this collected by questionnaire? This should be added to the methods.
Line 156- the authors state that sedentary behavior was measured. This is not correct since the pedometer measures step counts, not time spent in specific activity intensities. Therefore, this should be removed.
Line 173: ‘Enter Method’ appears to have been left in the manuscript. Please state the method that was used or remove.
Results
In line 163, ‘regular participation in OLTPA’ is defined as 3+ times/week, however in lines 181-183, the authors state that ‘regular’ participation estimates include those with 3+ times/week as well as those with 1-2 times/week. Please clarify what the definition of ‘regular’ participation is (I would argue that it’s 3+ times/week) and report sample information accordingly.
Figure 1: the italicized numbers for mothers and father are difficult to distinguish (vs. the youth numbers). I suggest revising with perhaps dashed lines around the mother/father boxes, or changing the shading pattern to make it easier to read. It’s also not clear what the dashed lines are for, since they are not identified in the legend.
Figure 2: the same comment as Figure 1 regarding the dashed lines
Figure 3: as these are all separate groups, the lines are misleading (making it appear like longitudinal data, which they are not). These prevalence estimates should be represented using bars (similar to Figure 2).
Lines 207-210: This sentence is very difficult to read. I think what the authors are trying to state is differences in obesity prevalence by sport participation in kids with overweight/obese parents? Suggest re-wording for clarity.
Throughout the results and in the figures, is it always the prevalence of obesity in youth that is estimated? Or do these numbers also include overweight youth, as is outlined in the Methods? Please clarify.
Lines 248-254: these sentences are awkwardly worded, making it difficult to understand what the authors are trying to say. I think they have flipped the X and Y in their interpretations, and for example the sentence should read ‘the odds of childhood obesity were lower if children achieve the recommended daily step count and were higher if mothers were overweight/obese’. Please review and clarify.
Discussion
Please review and edit for awkward wording throughout.
Lines 258, 267, 269 (and throughout the Discussion): ‘incidence’ is not correct, since this was not a longitudinal study studying how PA and OLTPA predicted obesity. ‘Prevalence’ is the more correct term since this analysis was cross-sectional and measures were taken at the same time. Please revise throughout the Discussion.
The authors are linking the participation in OLTPA with lower prevalence of obesity in youth. However, it could be that youth who are not obese are more likely to self-select into OLTPA. This should be added to the limitations.
Author Response
COVER LETTER – IJERPH-978621
5-November-2020
Dear Ms. Andjelka Jovanovic (Assistant Editor, MDPI),
Thank you very much for your comments and the opportunity to revise our manuscript. Please find below the List of Changes relating to the manuscript IJERPH-978621, titled "The relationship between obesity and physical activity of children in the spotlight of their parents' body weight" (previous title: "Can physical activity be a buffer for the prevalence of offspring obesity regardless of the parents' excessive body weight?"). We have provided point-by-point responses, each of them preceded by the respective reviewer’s comment. The changes made in the manuscript are highlighted in red-coloured text. In this way, all changes can easily be tracked in the manuscript.
In accordance with the Editor’s comment, we have added the data relating to the ethical approvals of the study to the modified version of the manuscript (part of the methodology).
The reviewers’ comments that we accepted and incorporated in the manuscript are marked in red-coloured text in the List of Changes.
Once again, we would like to thank you and all the reviewers and the editor for their stimulating comments, which were of great help in the process of revision.
Yours sincerely,
Erik Sigmund
On behalf of the co-authors
List of Changes
__________________________________________________________
Reviewer Report 1:
(Reviewer 1): This manuscript described the relationship between obesity, step counts, and presumably self-reported time spent in OLTPA in youth and parent dyads. Below are comments to improve the manuscript:
Introduction
The authors present at well-reviewed justification for their study. However, in places the wording is awkward, and should be edited for correct grammar and word choice.
Thank you for seeing the potential of our work. Thank you very much for this comment. In line with a similar comment by Reviewer 2, we have had the English checked and corrected in the introduction, as well as in the whole manuscript, by a native speaker.
Methods
Lines 125-132: were parents and children instructed to take anthropometric measurements fasting?
Yamax pedometer: how do the authors think knowledge of step count influenced their findings by providing direct participant feedback? Aside from removing the first day, were only whole-days included in analyses? Were there any criteria for number of whole days used? Also, later in the manuscript, the authors say they recommended a step count, however this procedure is not well-described. Was this an intervention study? Did they tell both the parents and children to increase their step counts? More information is needed about how this was communicated.
How was OLTPA measured? Was this collected by questionnaire? This should be added to the methods.
Thank you for pointing these issues out and your close attention. In line with similar comments by Reviewers 3 and 4, we have completely reformulated part of the Methods and added information on the type of study (cross-sectional), the selection of respondents, the number of schools from urban/rural locations, and the inclusion/exclusion criteria (added Scheme 1), a more detailed description of the way the research is organized, the % overweight and not only obesity (Table 1), and recording OLTPA using a questionnaire. Please see the reworked Methods section in the revised version of the manuscript.
Line 156- the authors state that sedentary behavior was measured. This is not correct since the pedometer measures step counts, not time spent in specific activity intensities. Therefore, this should be removed.
Thank you very much for this comment. We agree with your proposal and we have removed the words “and sedentary behaviour”.
Line 173: ‘Enter Method’ appears to have been left in the manuscript. Please state the method that was used or remove.
Binary logistic regression analyses were used in the study and the results are presented in Table 2.
Results
In line 163, ‘regular participation in OLTPA’ is defined as 3+ times/week, however in lines 181-183, the authors state that ‘regular’ participation estimates include those with 3+ times/week as well as those with 1-2 times/week. Please clarify what the definition of ‘regular’ participation is (I would argue that it’s 3+ times/week) and report sample information accordingly.
Thank you for this comment and your close attention. In reply to the comment, we have defined regular participation in OLTPA as 3+ times and added an appropriate quotation. The first lines (mentioned as lines 181-183) in the results section have been reworded regarding the definition of regular participation in OLTPA.
Figure 1: the italicized numbers for mothers and father are difficult to distinguish (vs. the youth numbers). I suggest revising with perhaps dashed lines around the mother/father boxes, or changing the shading pattern to make it easier to read. It’s also not clear what the dashed lines are for, since they are not identified in the legend.
Figure 2: the same comment as Figure 1 regarding the dashed lines
Thank you for pointing these issues out and your close attention. In line with a similar comment by Reviewers 1 and 4, we have redesigned Figures 1 and 2 to make the categories of family members clearer (using bars and colour-coding categories).
Figure 3: as these are all separate groups, the lines are misleading (making it appear like longitudinal data, which they are not). These prevalence estimates should be represented using bars (similar to Figure 2).
Thank you very much for this comment. We agree with your proposal and we have changed the format of Figure 3 and used the bars and colour differentiation of categories.
Lines 207-210: This sentence is very difficult to read. I think what the authors are trying to state is differences in obesity prevalence by sport participation in kids with overweight/obese parents? Suggest re-wording for clarity.
Many thanks for this comment. We have reformulated the sentence for greater clarity.
Throughout the results and in the figures, is it always the prevalence of obesity in youth that is estimated? Or do these numbers also include overweight youth, as is outlined in the Methods? Please clarify.
Thank you very much for this comment. Throughout the results and figures, the prevalence of obesity in children is used, in accordance with the formulated goals and hypothesis of the study.
Lines 248-254: these sentences are awkwardly worded, making it difficult to understand what the authors are trying to say. I think they have flipped the X and Y in their interpretations, and for example the sentence should read ‘the odds of childhood obesity were lower if children achieve the recommended daily step count and were higher if mothers were overweight/obese’. Please review and clarify.
Many thanks for this comment. We agree with your proposal and we have reformulated the part of the Results section that you mentioned.
Discussion
Please review and edit for awkward wording throughout.
Lines 258, 267, 269 (and throughout the Discussion): ‘incidence’ is not correct, since this was not a longitudinal study studying how PA and OLTPA predicted obesity. ‘Prevalence’ is the more correct term since this analysis was cross-sectional and measures were taken at the same time. Please revise throughout the Discussion.
Thank you for this comment and your close attention. This was our inaccuracy. In line with this comment, we have thoroughly checked and replaced the term ‘incidence’ with the term ‘prevalence’.
The authors are linking the participation in OLTPA with lower prevalence of obesity in youth. However, it could be that youth who are not obese are more likely to self-select into OLTPA. This should be added to the limitations.
You are right. In accordance with this comment, we have added this consideration to the limitations of this study too.

Reviewer 2 Report
Thank you submitting your manuscript: “Can physical activity be a buffer for the prevalence of offspring obesity regardless of the parents' excessive body weight?” I hope you find the following comments constructive:
Throughout the manuscript please ensure abbreviations are correctly capitalized, e.g. Physical Activity (PA); Organized Leisure-Time PA (OLTPA) etc.
Introduction reads well overall.
Line 61 “but rarely with regard to children's participation in OLTPA” although rarely, there is some research, please provide the citations here.
Line 64 “and its incidence cannot be satisfactorily reduced globally [25,26]” , please amend, to has not been (cannot implies it never will be).
Line 66 insert the word often.. “ that OFTEN results in excessive caloric intake, as well as insufficient PA “
Line 81-82 “Since the participation of children and adolescents in OLTPA in Czechia is the most widespread form of spending organized leisure time in young people (≥ 74% for boys and ≥ 63% for girls [5,31])” This is a difficult sentence for a reader to grasp, I understand but I recommend you restructure the sentence.
Amend language “this paper” line 86, no the paper didn’t sort anything.
Aims and justification clear, but please clearly state the study hypothesis after the aims.
Please refer to the hypothesis directly in the results and followed-up in the discussion section.
The article is well written. Minor amends only.
Author Response
COVER LETTER – IJERPH-978621
5-November-2020
Dear Ms. Andjelka Jovanovic (Assistant Editor, MDPI),
Thank you very much for your comments and the opportunity to revise our manuscript. Please find below the List of Changes relating to the manuscript IJERPH-978621, titled "The relationship between obesity and physical activity of children in the spotlight of their parents' body weight" (previous title: "Can physical activity be a buffer for the prevalence of offspring obesity regardless of the parents' excessive body weight?"). We have provided point-by-point responses, each of them preceded by the respective reviewer’s comment. The changes made in the manuscript are highlighted in red-coloured text. In this way, all changes can easily be tracked in the manuscript.
In accordance with the Editor’s comment, we have added the data relating to the ethical approvals of the study to the modified version of the manuscript (part of the methodology).
The reviewers’ comments that we accepted and incorporated in the manuscript are marked in red-coloured text in the List of Changes.
Once again, we would like to thank you and all the reviewers and the editor for their stimulating comments, which were of great help in the process of revision.
Yours sincerely,
Erik Sigmund
On behalf of the co-authors
List of Changes
__________________________________________________________
Reviewer Report 2:
(Reviewer 2): Thank you submitting your manuscript: “Can physical activity be a buffer for the prevalence of offspring obesity regardless of the parents' excessive body weight?” I hope you find the following comments constructive:
Thank you for seeing the potential of our work.
Throughout the manuscript please ensure abbreviations are correctly capitalized, e.g. Physical Activity (PA); Organized Leisure-Time PA (OLTPA) etc.
Thank you for this comment and your close attention. This was our inaccuracy. In line with this comment, we have thoroughly checked the writing of abbreviations in capital letters throughout the manuscript.
Introduction reads well overall.
Line 61 “but rarely with regard to children's participation in OLTPA” although rarely, there is some research, please provide the citations here.
Thank you very much for this comment. We do agree with it, and in line with it, we have added a suitable quote [23].
Line 64 “and its incidence cannot be satisfactorily reduced globally [25,26]”, please amend, to has not been (cannot implies it never will be).
Line 66 insert the word often. “that OFTEN results in excessive caloric intake, as well as insufficient PA“
Thank you very much for this comment. This was our inaccuracy. We completely agree with your proposal and we have amended/inserted the terms you mentioned.
Line 81-82 “Since the participation of children and adolescents in OLTPA in Czechia is the most widespread form of spending organized leisure time in young people (≥ 74% for boys and ≥ 63% for girls [5,31])” This is a difficult sentence for a reader to grasp, I understand but I recommend you restructure the sentence.
Many thanks for these comments. You are right that the original passage of the text was not reader-friendly. In accordance with these comments, we have rephrased these sentences.
Amend language “this paper” line 86, no the paper didn’t sort anything.
Thank you for this comment and your close attention. This was our inaccuracy. Following this recommendation, we have reformulated the sentence you mention.
Aims and justification clear, but please clearly state the study hypothesis after the aims. Please refer to the hypothesis directly in the results and followed-up in the discussion section.
Thank you very much for this comment. We agree with your proposal and we have added a hypothesis immediately after the formulated aims. “In this study, we will test the hypothesis whether there is a difference in the prevalence of obesity in offspring without and with regular participation in OLTPA (three or more times per week) in the case of excessive body weight in their parents.”

Reviewer 3 Report
Dear Authors,
The following comments may strengthen the manuscript:
Abstract
Please indicate the country of the observation.
Introduction
lines 41-52: The authors discussed various mechanisms of parents' influence on their children's physical activity. They also mentioned an interesting relationship between the greater influence of the same-sex parent. Could the authors here also refer to the possible influence of parents of children of different ages? Is this influence of parents still observed among adolescents (the study concerns a group aged 4-16)?
Materials and methods
I understand that the detailed research methodology was described earlier. However, for the convenience of the reader and better understanding, it is advisable to add a survey diagram or a more detailed description as supplementory materials/appendix to the present article.
Table 1. In this table only the obesity prevelance in study group is included. But in abstract authors also refer to overweight (lines 19-20), so in my opinion numbers for overweight rate shoud be included in the table as well.
Results
Figure 1. The method of marking the respondents (daughters, sons, mothers, fathers) requires changing because it is illegible. Please use different mark for each group. Instead of the arrow, I suggest using the standard significance marks (*).
Figure 2. Instead of the arrow, I suggest using the standard significance mark (e.g. different characters/letters).
Table 2: In the table, both the criterion of excess body weight (overweight / obesity) and only obesity were used. This is confusing to the reader.
Discussion
This part is well led. However, in my opinion, there is no analysis of dependencies within diffrent age groups (also in the results). The 4-16 age range is very wide, the influence of parents on the child's behavior is undoubtedly different among preschool / early school children compared to adolescents. Could the authors relate to this?
Author Response
COVER LETTER – IJERPH-978621
5-November-2020
Dear Ms. Andjelka Jovanovic (Assistant Editor, MDPI),
Thank you very much for your comments and the opportunity to revise our manuscript. Please find below the List of Changes relating to the manuscript IJERPH-978621, titled "The relationship between obesity and physical activity of children in the spotlight of their parents' body weight" (previous title: "Can physical activity be a buffer for the prevalence of offspring obesity regardless of the parents' excessive body weight?"). We have provided point-by-point responses, each of them preceded by the respective reviewer’s comment. The changes made in the manuscript are highlighted in red-coloured text. In this way, all changes can easily be tracked in the manuscript.
In accordance with the Editor’s comment, we have added the data relating to the ethical approvals of the study to the modified version of the manuscript (part of the methodology).
The reviewers’ comments that we accepted and incorporated in the manuscript are marked in red-coloured text in the List of Changes.
Once again, we would like to thank you and all the reviewers and the editor for their stimulating comments, which were of great help in the process of revision.
Yours sincerely,
Erik Sigmund
On behalf of the co-authors
List of Changes
__________________________________________________________
Reviewer Report 3:
(Reviewer 3): Dear Authors, The following comments may strengthen the manuscript:
Thank you for seeing the potential of our work.
Abstract
Please indicate the country of the observation.
Thank you very much for this comment. We do agree with this comment. In line with it, we have added the country where the research was conducted.
lines 41-52: The authors discussed various mechanisms of parents' influence on their children's physical activity. They also mentioned an interesting relationship between the greater influence of the same-sex parent. Could the authors here also refer to the possible influence of parents of children of different ages? Is this influence of parents still observed among adolescents (the study concerns a group aged 4-16)?
Thank you for this comment. In reply to the comment, we have extended the paragraph you mention with other examples and we have also added the possible influence of parents on children in different age groups.
Materials and methods
I understand that the detailed research methodology was described earlier. However, for the convenience of the reader and better understanding, it is advisable to add a survey diagram or a more detailed description as supplementary materials/appendix to the present article.
Thank you for pointing these issues out and your close attention. In line with similar comments by Reviewers 1 and 4, we have completely reformulated part of the Methods and added information on the type of study (cross-sectional), the selection of respondents, the number of schools from urban/rural locations, and the inclusion/exclusion criteria (added Scheme 1), a more detailed description of the way the research is organized, the % overweight, and not only obesity (Table 1), and recording OLTPA using a questionnaire. Please see the reworked Methods section in the revised version of the manuscript.
Table 1. In this table only the obesity prevalence in study group is included. But in abstract authors also refer to overweight (lines 19-20), so in my opinion numbers for overweight rate should be included in the table as well.
Thank you for this comment and your close attention. We agree with your proposal and we have added the prevalence (%) of overweight participants in Table 1.
Results
Figure 1. The method of marking the respondents (daughters, sons, mothers, fathers) requires changing because it is illegible. Please use different mark for each group. Instead of the arrow, I suggest using the standard significance marks (*).
Thank you for pointing these issues out and your close attention. In line with a similar comment by Reviewers 1 and 4, we have redesigned Figure 1 to make the categories of family members clearer (using bars and colour-coding categories).
Figure 2. Instead of the arrow, I suggest using the standard significance mark (e.g. different characters/letters).
Thank you very much for this comment. We agree with your proposal and we have changed the format for emphasizing significance.
Table 2: In the table, both the criterion of excess body weight (overweight / obesity) and only obesity were used. This is confusing to the reader.
Thank you for this comment and your close attention. It may seem confusing to the reader, but this is in line with the required formulation of the hypothesis. We have emphasized this further in the text of the results.
Discussion
This part is well led. However, in my opinion, there is no analysis of dependencies within diffrent age groups (also in the results). The 4-16 age range is very wide, the influence of parents on the child's behavior is undoubtedly different among preschool / early school children compared to adolescents. Could the authors relate to this?
Thank you for pointing these issues out and your close attention. To reflect on the comment, we also performed regression analyses with respect to the age categories of the children, but no differences were found in the results with respect to the age categories of the children. We have emphasized this in the legend of Table 2 and in the text of the results.

Reviewer 4 Report
1) Please provide more detail about how many days/weeks over the study period were observed.
2) There are 915 mother-child dyads and 578 father-child dyads in the data, but it is not well reflected in Table 1. It would be more informative if descriptive statistics of the participants can be summarized in Table 1 by the type of dyad as in Table 2 (Mother-daughter; Father-daughter; Mother-son; Father-son). If possible, please summarize data by both the type of dyad and the OLTPA participation status.
3) I wonder if there is any other factor measured in data. No covariate was controlled for in logistic regression. This is critical since data is cross-sectional.
4) It is very hard to talk about the cause and effect with cross-sectional data. Please consider revising the title and the text throughout the text.
5) It is not clear how the effects of four types of dyad in data (Mother-daughter; Father-daughter; Mother-son; Father-son) can vary. For example, a mother-daughter dyad exists in data not because there was only mother in the family. It is simply because the mother provided the informed consent form. Information about father is missing and that is one major limitation of the study. If this is correct, then the author(s) need to provide more persuasive reasons why the data needs to be analyzed by father/mother-son/daughter dyad, not parent-child dyad.
6) Please provide a little bit more detail about the randomization process. For example, there is no information about how many schools are selected considering GDP in the 102 administrative regions (out of how many administrative regions?) in Czech. And please also provide some information about how urban and rural areas were defined prior to randomization and how many regions in each category.
7) Figure 1 and 2 would make more sense if data were displayed by the weight status of mother or father considering the main research question. Still, Figure 1 seems to be not related to the main research question (child obesity and parent’s weight status).
8) Figure 1 (step counts) and 2 (obesity) are descriptive results at best. The key to answer the main research question is from the logistic regression results. However, the main conclusion, “To conclude, those children with regular participation in OLTPA (three or more times a week) had a significantly lower prevalence of obesity than those children without participation in OLTPA”, is not really supported by the logistic regression results in Table 2.
9) Please provide the sample size for each combination (Mother-daughter; Father-daughter; Mother-son; Father-son) in Table 2.
10) Please consider an appropriate analysis method that can handle the clustering effect at the school level. Further, please also consider a modeling approach designed for dyad data for better analysis: http://www.davidakenny.net/dyad.htm
11) The definition of “obesity” is described in text later, but to avoid any confusion, please specify “obesity” in Table 1 or as a note.
12) Please include the term “OLTPA” in Table 2 for consistency throughout the text.
13) In Figure 1, please change the colors of the legends. It is very hard to differentiate daughters from mothers (even with italic) and sons from fathers in black/white.
14) Figure 1 could be more informative if presented in a table with numbers.
Author Response
COVER LETTER – IJERPH-978621
5-November-2020
Dear Ms. Andjelka Jovanovic (Assistant Editor, MDPI),
Thank you very much for your comments and the opportunity to revise our manuscript. Please find below the List of Changes relating to the manuscript IJERPH-978621, titled "The relationship between obesity and physical activity of children in the spotlight of their parents' body weight" (previous title: "Can physical activity be a buffer for the prevalence of offspring obesity regardless of the parents' excessive body weight?"). We have provided point-by-point responses, each of them preceded by the respective reviewer’s comment. The changes made in the manuscript are highlighted in red-coloured text. In this way, all changes can easily be tracked in the manuscript.
In accordance with the Editor’s comment, we have added the data relating to the ethical approvals of the study to the modified version of the manuscript (part of the methodology).
The reviewers’ comments that we accepted and incorporated in the manuscript are marked in red-coloured text in the List of Changes.
Once again, we would like to thank you and all the reviewers and the editor for their stimulating comments, which were of great help in the process of revision.
Yours sincerely,
Erik Sigmund
On behalf of the co-authors
List of Changes
__________________________________________________________
Reviewer Report 4:
(Reviewer 4):
1) Please provide more detail about how many days/weeks over the study period were observed.
2) There are 915 mother-child dyads and 578 father-child dyads in the data, but it is not well reflected in Table 1. It would be more informative if descriptive statistics of the participants can be summarized in Table 1 by the type of dyad as in Table 2 (Mother-daughter; Father-daughter; Mother-son; Father-son). If possible, please summarize data by both the type of dyad and the OLTPA participation status.
6) Please provide a little bit more detail about the randomization process. For example, there is no information about how many schools are selected considering GDP in the 102 administrative regions (out of how many administrative regions?) in Czech. And please also provide some information about how urban and rural areas were defined prior to randomization and how many regions in each category.
Thank you for pointing these issues out and your close attention. In line with similar comments by Reviewers 1 and 3, we have completely reformulated part of the Methods and added information on the type of study (cross-sectional), the selection of respondents, the number of schools from urban/rural locations, the number of days that were monitored, and the inclusion/exclusion criteria (added Scheme 1), a more detailed description of the way the research is organized, the % overweight and not only obesity (Table 1), and recording OLTPA using a questionnaire. Please see the reworked Methods section in the revised version of the manuscript.
3) I wonder if there is any other factor measured in data. No covariate was controlled for in logistic regression. This is critical since data is cross-sectional.
Thank you very much for this comment. The study also monitored the screen time and, in more detail, the PA on each day of the week and the number of siblings (and their PA and screen time). However, these variables are the subject of another manuscript. Because of the non-overlapping topics, they were not included here.
4) It is very hard to talk about the cause and effect with cross-sectional data. Please consider revising the title and the text throughout the text.
Thank you for pointing this issue out. We agree with your proposal and we have changed the tittle of the study and modified the wording regarding causes or effects throughout the manuscript.
5) It is not clear how the effects of four types of dyad in data (Mother-daughter; Father-daughter; Mother-son; Father-son) can vary. For example, a mother-daughter dyad exists in data not because there was only mother in the family. It is simply because the mother provided the informed consent form. Information about father is missing and that is one major limitation of the study. If this is correct, then the author(s) need to provide more persuasive reasons why the data needs to be analyzed by father/mother-son/daughter dyad, not parent-child dyad.
Thank you for pointing this issue out. To reflect on the comment, in the reformulated part of the methodology, we stated that all family members were monitored and all the dyads that met the inclusion criteria were included in the subsequent analyses. Previous studies show that there are differences in the relationships between the PA of parents and of their children with respect to their gender, and so we performed analyses with respect to the sex of the participants. We have also added a justification to Section 2.4. Data Processing and Statistical Analysis.
7) Figure 1 and 2 would make more sense if data were displayed by the weight status of mother or father considering the main research question. Still, Figure 1 seems to be not related to the main research question (child obesity and parent’s weight status). 13) In Figure 1, please change the colors of the legends. It is very hard to differentiate daughters from mothers (even with italic) and sons from fathers in black/white. 14) Figure 1 could be more informative if presented in a table with numbers.
Many thanks for these comments. Given the positive comments of Reviewers 1 and 3, we left the descriptive Figures 1 and 2, but changed their design to make the categories of family members clearer.
8) Figure 1 (step counts) and 2 (obesity) are descriptive results at best. The key to answer the main research question is from the logistic regression results. However, the main conclusion, “To conclude, those children with regular participation in OLTPA (three or more times a week) had a significantly lower prevalence of obesity than those children without participation in OLTPA”, is not really supported by the logistic regression results in Table 2.
Many thanks for these comments. You are right. In accordance with these comments, we have reformulated the sentences in the results and conclusions and modified our causal statement.
9) Please provide the sample size for each combination (Mother-daughter; Father-daughter; Mother-son; Father-son) in Table 2.
Thank you very much for this comment. In line with this comment, we have added a sample size for each combination (mother-daughter, father-daughter, mother-son, father-son) in Table 2.
10) Please consider an appropriate analysis method that can handle the clustering effect at the school level. Further, please also consider a modelling approach designed for dyad data for better analysis: http://www.davidakenny.net/dyad.htm
Thank you for this comment and your close attention. This was our inaccuracy. We performed a cluster analysis, but this analysis did not reveal PA indicators for clustering by school. We have added a mention of this to Section 2.4. Data Processing and Statistical Analysis. Thank you very much for the approach to dyad data modelling that you monitored, which we will use to create the future manuscript of another article in the future.
11) The definition of “obesity” is described in text later, but to avoid any confusion, please specify “obesity” in Table 1 or as a note.
Thank you very much for this comment. We do agree with it, and in line with it, we have added the definition of overweight/obesity to the legend of Table 1.
12) Please include the term “OLTPA” in Table 2 for consistency throughout the text.
Thank you for this comment and your close attention. This was our inaccuracy. In line with this comment, we have included the abbreviation OLTPA in Table 2.

Round 2
Reviewer 4 Report
The authors claim that "The key finding of the study is the confirmation of the widespread regular participation of Czech children in OLTPA, which, together with other daily PA, was positively associated with a lower prevalence of childhood obesity even in the case of excessive weight of their parents."
However, the logistic regression results show that
1) participation of children in OLTPA is not really associated with a lower or higher likelihood of obesity among sons or daughters, except the father-son dyad
2) children, regardless of gender, are more likely to be obese when his or her mother is overweight/obese. Further, this is true even after controlling for daily SC and OLTPA participation of the child and the mother.
And the interaction between child's OLTPA and weight status of parent is not tested so the logistic regression results do not show that "...was positively associated with a lower prevalence of childhood obesity even in the case of excessive weight of their parents."
The conclusions are based on descriptive results in Figure 1 and 2, so not relevant.
Author Response
Reviewer Report 4:
(Reviewer 4):
The authors claim that "The key finding of the study is the confirmation of the widespread regular participation of Czech children in OLTPA, which, together with other daily PA, was positively associated with a lower prevalence of childhood obesity even in the case of excessive weight of their parents."
However, the logistic regression results show that
1) participation of children in OLTPA is not really associated with a lower or higher likelihood of obesity among sons or daughters, except the father-son dyad
2) children, regardless of gender, are more likely to be obese when his or her mother is overweight/obese. Further, this is true even after controlling for daily SC and OLTPA participation of the child and the mother.
And the interaction between child's OLTPA and weight status of parent is not tested so the logistic regression results do not show that "...was positively associated with a lower prevalence of childhood obesity even in the case of excessive weight of their parents."
The conclusions are based on descriptive results in Figure 1 and 2, so not relevant.
Thank you for pointing these issues out and your close attention. We agree with your proposal and we have reformulated the first part of the Discussion section and the whole Conclusions section.
ORIGINAL TEXT:
- Discussion
The key finding of the study is the confirmation of the widespread regular participation of Czech children in OLTPA, which, together with other daily PA, was positively associated with a lower prevalence of childhood obesity even in the case of excessive weight of their parents.
CHANGED TEXT:
- Discussion
The key finding of the study is the confirmation of the hypothesis concerning the significant difference in the prevalence of obesity in offspring with and without and with regular participation in OLTPA (three or more times per week) in the case of excessive body weight in their parents. However, regular participation of children in OLTPA is not significantly associated with a lower likelihood of obesity among sons or daughters, except for father-son dyads. In the case of sons, their likelihood of obesity is reduced by reaching the recommended daily SC, regardless of mother/father-son dyads. Children, regardless of gender, are more likely to be obese if their mother is overweight/obese.
ORIGINAL TEXT:
- Conclusions
Even though causality cannot be established, the cumulative effect of regular participation in OLTPA and children's own PA seems to be a stronger alleviator of children's obesity than the risk posed by the excessive body weight of their parents. Children with regular participation in OLTPA (three or more times a week) had a significantly lower prevalence of obesity than those children without participation in OLTPA. Even in the case of parents (mothers/fathers) with excessive body weight, children with regular participation in OLTPA had a significantly lower prevalence of obesity than children without OLTPA. The mother’s overweight/obesity significantly increases the odds of her offspring being overweight/obese. However, achieving the recommended daily SC and regular participation in OLTPA significantly reduces the chances of obesity, especially in sons.
CHANGED TEXT:
- Conclusions
The lowest prevalence of obesity was observed in offspring with regular participation in OLTPA (three or more times per week) compared with offspring without participation in OLTPA, even in the case of excessive body weight in their parents. However, regular participation of children in OLTPA was not significantly associated with a lower likelihood of obesity among sons or daughters, except for father-son dyads. Children, regardless of gender, were more likely to be obese if their mothers were overweight/obese. It turns out that a satisfactory explanation for the prevalence of childhood obesity should include other energy-related behaviour, lifestyle and environmental variables in addition to anthropometric and PA data of parents.
